# How Is Work Ability Shaped in Groups of Shift and Non-Shift Workers? A Comprehensive Approach to Job Resources and Mediation Role of Emotions at Work

**DOI:** 10.3390/ijerph18157730

**Published:** 2021-07-21

**Authors:** Łukasz Baka, Dawid Ścigała, Łukasz Kapica, Andrzej Najmiec, Krzysztof Grala

**Affiliations:** 1Laboratory of Psychology and Sociology of Work, Central Institute for Labour Protection—National Research Institute, 00-701 Warsaw, Poland; lukap@ciop.pl (Ł.K.); annaj@ciop.pl (A.N.); 2Institute of Psychology, The Maria Grzegorzewska University of Warsaw, 02-353 Warsaw, Poland; kgSD6@aps.edu.pl

**Keywords:** work ability, shift work schedule, job resources, emotions

## Abstract

There is much less research on the relationship between shift work and positive states experienced in the workplace, e.g., emotions and work ability. Using the job demands-resources model, conservation of resources theory and the broaden-and-build theory as theoretical frameworks, the direct and indirect (mediated via positive and negative emotions) relations between the complex of job resources and work ability were tested in the group of shift and non-shift workers. Three types of resources related to task, leadership and interpersonal relations were taken into account. Data were collected among 1510 workers. Structural equation modelling (SEM) showed that only leadership was directly related to high work ability in both occupational groups. Indirect effects of positive and negative emotions were strongly supported. Both of them mediate the effects of the three analysed job resources on work ability, but individual job resources impacted were different. Leadership resources led to “good” work ability by the intensification of positive and reduction of negative emotions. Interpersonal resources decrease negative emotions. Task resources, conversely, intensified positive emotions, which in turn increased work ability. These research results shed more light on the ways of shaping work ability among shift and non-shift workers.

## 1. Introduction

In previous studies on shift workers’ physical and mental health, scientists have focused mainly on assessing the negative consequences of shift work. In several meta-analyses, they showed that shift work was related to depression [1,2,3], strain [4], unhealthy eating [5], overweight and obesity [6,7], insomnia [8,9], as well as high risk of cardiovascular disease [10,11,12], stroke [13] and cancers [14,15,16]. Relatively little attention, however, has been devoted to positive phenomena related to work, e.g., work ability. Therefore, there is a need to identify not only the possible causes of decreased wellbeing of shift workers but also the factors that may actually improve it. In the current study, we investigate whether and how different types of job-related psychosocial factors and emotions at work shape “high” work ability in groups of shift and non-shift workers. Specifically, we try to answer the following questions: what types of job resources are the most conducive to the development of high work ability among shift and non-shift workers, as well as whether positive and negative emotions at work mediate the job resources–work ability link.

### 1.1. The Theoretical Framework of the Current Research

The theoretical framework for this research is provided by the job demands-resources (JD-R) model [17], conservation of resources (COR) theory [18] and broaden-and-build (B&B) theory [19,20]. JD-R model states that demands and resources affect people through dual processes—energetic and motivational. In the first of these processes, prolonged job demands result in depleting job resources and job burnout, which in the long run leads to poor mental health. According to the second process, more important from the perspective of the research goal undertaken, job resources favour the development of a positive state (e.g., work engagement), which leads to desirable organisational outcomes such as commitment [21] and job performance [22]. To the best of our knowledge, there are five studies devoted to the relations between job resources and work ability [23,24,25,26,27]—neither of these concerned shift workers. Additionally, a limited number of job resources were included in the studies, and they are seldom operationalised by the means of single variables, e.g., supervisory and co-workers’ relations [24], skill discretion [25], opportunities for development [26], as well as autonomy and strengths [27]. However, as COR theory assumes, it is unlikely that resources exist in isolation of each other [28]. People try to increase the pool of their own resources (e.g., by accumulating or investing them), so there is a positive gain spiral between resources—different types of resources interact and multiply [29,30]. Therefore, they should be studied comprehensively. Based on the typology of Berthelsen et al. [31], we propose a complete approach to job resources, including 13 types of them, classified into three groups—task, leadership and interpersonal resources [32]—in the current study. It will be investigated which of these three types of job resources is the most conducive to the development of work ability in shift and non-shift workers groups.

The role of negative emotions at work is well documented [33]. A little less research is devoted to the consequences of positive emotions. According to the B&B theory, positive emotions play a key role in building resources and enhancing employees’ mental health [19,20]. Their mediation function was supported for several job-related outcomes—e.g., job satisfaction [34], work engagement [35], organisational citizenship behaviour [36]—but not for work ability. In addition to the direct effect of job resources on work ability, the mediational effects of positive and negative emotions among shift and non-shift workers will be tested in the current study. Specifically, we investigate whether having high job resources leads to the weakening of negative emotions or whether it can increase positive emotions at work. By combining the comprehensive approach to resources and job-related emotions, our research model integrates COR and B&B theories with the motivational process of the JD-R model.

### 1.2. Work Ability in Shift and Non-Shift Work

Over the past few years, workplace wellbeing has become a popular subject of research in work psychology [37,38]. Previously, the vast majority of studies focused on stressors, sources of dysfunctional behaviour and organisational pathology. A review of articles published in leading American journals in the field of work psychology at the turn of 1996–2004 [39] showed that 94% of them refer to the phenomena that Bakker and Schaufeli [40] included in the so-called Groups of four Ds—damages, diseases, disorders, and dysfunctions. Identifying the source of these phenomena and preventing them rarely results in an increase in wellbeing. This was noted almost half a century ago by Herzberg [41] in the two-factor theory when he wrote that positive organisational outcomes could be obtained by influencing motivators (e.g., growth) and not by improving hygiene factors (e.g., “poor” job conditions). This assumption has initiated the development of more modern psychological concepts, e.g., JD-R model [17] and B&B theory [20], which postulate that the health and wellbeing of employees (e.g., work ability) is shaped mainly by building resources (and not by minimising the hazardous factors).

According to the concept developed in 1980 by the Finnish Institute Occupational Health (FIOH), work ability refers to how well the workers are able to perform their work and was defined as a balance between the employee’s ability to perform work and the demands posed by work at every stage of working life [42,43]. In this concept, work ability is the results of interaction between the person, the task and the working environment. This means that work ability should not be determined without a detailed assessment of the type of work, job-related tasks and working environment [44]. Work ability is believed to be shaped by four groups of factors, including health (physical, mental and social), competencies (e.g., knowledge and skills), values (e.g., work motives and work attitudes) and environment (e.g., workplace, kind of work, demands). FIOH developed the work ability index (WAI) to measure workers’ perceptions regarding their physical, mental and social health and their ability to cope with job demands.

Research on antecedents of work ability is quite extensive; however, most studies thus far have focused on factors that might undermine work ability, e.g., job demands, poor job conditions, types of work (physical vs. mental), long work hours (e.g., [45,46,47,48,49]). For instance, a meta-analysis based on 14 cross-sectional studies and six longitudinal studies found that both individual factors (such as age or overweight) and work-related factors (such as high mental and physical work demands, low autonomy and poor ergonomic conditions) are related to low work ability [49]. A few studies investigated whether a shift-work schedule is critical for one’s work ability [50,51,52,53,54]. Two of them compare work ability in the shift and non-shift groups of workers [51,54]. Their results are inconsistent. For instance, Sorić et al. [54], in a study on 1124 nurses, found no significant differences in the level of work ability among nurses working in daytime and shift modes. Costa [51], on the other hand, showed that work ability worsens with age, with this effect being stronger in the group of shift workers compared to daytime workers. In a longitudinal study, Camerino [50] found that the work schedule was not related to changes in WAI after one year.

### 1.3. A Comprehensive Approach to Job Resources

Job resources refer to those physical, psychological, social, or organisational aspects of the job that reduce job demands and the associated physiological and psychological costs, help to achieve work goals and stimulate personal growth, learning and development [55]. Bakker and Demerouti [32] have proposed a division between job resources arising from task, leadership and interpersonal levels. This theoretical approach was supported empirically [31]. Using the Copenhagen Psychosocial Questionnaire (COPSOQ II), covering a broad range of aspects of the psychosocial working environment, Berthelsen et al. [31]—by means of confirmatory factor analysis—distinguished three job resource groups related to the task, leadership and interpersonal relations. Task resources include four factors related to influence at work, possibilities for development, work variation and role clarity. Interpersonal resources include three factors refer co-workers support, the social community at work and co-workers’ trust. Leadership resources, in turn, consist of six variables related to the quality of leadership, superior support, rewards, predictability, organisational justice and superior trust [31]. This typology was used in the current study.

Each of the three types of job resources may trigger a motivational process [32] and lead to high work ability because all of them satisfy three basic psychological needs—autonomy, competence and relatedness [56]. Task resources satisfy the need for autonomy (e.g., increasing opportunities to make decisions concerning work time or work tasks) and the need for competence (e.g., increasing opportunities to use their skills at work). Interpersonal resources satisfy the need for relatedness (e.g., receiving help from co-workers or strengthening social ties). Leadership resources, in turn, satisfy both the need for relatedness (e.g., support from supervisors) and the need for competence (e.g., through quality and frequency of receiving feedback from supervisor).

Based on COR theory [18,28], we suppose that task, interpersonal and leadership resources will have an additive effect on work ability. According to COR theory, resources are things that workers value, and therefore they must invest them to protect against future resource loss, recover their resources and gain new resources [18,28]. In other words, various job resources are salient factors in gaining new resources and enhancing wellbeing. Additionally, the COR theory posits that whereas those with fewer resources are more vulnerable to resource loss, those with greater resources are, correspondingly, less vulnerable to resource loss and more capable of resource gain [28]. Hence, gaining resources increases the resource pool, making it more likely that additional resources will be subsequently acquired. According to COR theory, this accumulation and linking of resources create “resource caravans”. That is, resources tend not to exist in isolation, but instead, they multiply each other, creating the so-called positive gain spiral. For example, high predictability at work and high organisational justice (leadership resources) will likely foster higher feelings of influence and job control (task resources), and in turn, these will translate into more intensive activities in seeking social support and stronger relationships with other employees (interpersonal resources). Several studies confirmed that in the long run, such resource caravans result in positive personal outcomes like better coping, adaptation and wellbeing [29,30,57].

Three cross-sectional [25,26,27] and two cross-lagged studies [23,24] confirm job resources–work ability link. For example, “good” relations with a supervisor were predictors of high work ability, measured ten years later in a group of Finnish firefighters [24]. In a similar vein, work ability was related to high autonomy [26], skill discretion [25] and strengths [27]. A limitation of these studies is that quite a small number of resources that were considered. We propose a more comprehensive approach to job resources, including 13 types categorised into three job resources groups—task, interpersonal and leadership resources. Positive relations between the three groups of job resources and work ability are hypothesised.

### 1.4. Mediation Role of Positive and Negative Emotions at Work

Although initially researchers perceived positive and negative affects as two opposing theoretical constructs [58], they are now treated as two relatively autonomous and unipolar dimensions, moderately correlated with each other [59]. Whereas positive affect reflects the extent to which individuals generally feel active, excited, enthusiastic, inspired and proud, negative affect reflects the extent to which individuals generally feel upset or unpleasantly aroused [60]. Work-related emotions are described as relatively intense, short-lived affective experiences that are focused on specific objects or situations at work [61]. Emotions not only make people feel good or bad at a particular point in time, but they may also produce future wellbeing at work [62].

According to the B&B theory, positive emotions broaden thought–action repertoires by inducing exploratory behaviours that create learning opportunities and goal achievement [19,20]. In short, positive emotions make people grow. Many positive emotions have a social origin. This means that social interactions in the workplace may play an important role in their formation [63]. For example, being popular or liked with a group of co-workers can be a source of joy and contentment. In turn, positive feedback from the supervisor may be a reason for enthusiasm or pride in the work done. This is confirmed by research. For instance, Schaufeli and van Rhenen [64] found that employees working in resourceful work environments, characterised by quality feedback, learning opportunities, and autonomy, often feel enthusiasm, pride, and joy while working. A diary study conducted over two weeks showed that employees positively assessed the quality of leadership experience more positive emotions throughout the workday [65]. However, several studies found that positive emotions may be triggered by job resources [66,67,68] and lead to wellbeing at work, e.g., work engagement [35] or job satisfaction [34], neither of them concerned work ability. However, Fredrickson [69] suggests that positive emotions foster resilience, build work competencies and improve health, which—as stated above—are critical factors for the shaping of work ability. For instance, some researchers showed that positive affect is a predictor of lower blood fat, blood pressure and a healthier body mass index [70], lower risk for heart disease [71] and improved self-rated health [72]. Positive emotions also serve a health-protective function after negative experiences. An experimental study found that inducing positive feelings led to faster cardiovascular recovery [73]. On the basis of the above studies, we hypothesised that the three types of job resources trigger positive emotions at work, which in turn lead to high work ability.

Even though positive emotions happen more frequently at work, negative ones are more easily recalled and have a stronger effect on overall affective outcomes at work [74]. Several studies found that negative emotions are related to “poor” physical and mental health, including coronary heart disease [75], chronic physical illness [76], insomnia [77], anxiety and depression [78], which may be treated as antecedents of work disability [42,46]. Moreover, increasing positive emotions and high job resources can also weaken negative emotional states at the workplace [79]. Therefore, we also introduced a negative affect to our study. We perceive positive and negative emotions as two potential processes (mediators) through which job resources shape “good” work ability in shift and non-shift workers groups. We also compare which of the emotions more strongly mediate beneficial effects of job resources on work ability. In other words, we try to investigate whether job resources lead to stronger work ability by intensifying positive emotions or by weakening negative ones. Based on the above considerations, a theoretical model was created, which is presented in Figure 1.

## 2. Materials and Methods

### 2.1. Participants and Procedures

The sample study *(n* = 1510) includes Polish shift (*n* = 760; 50.3%) and non-shift (*n* = 750; 49.7%) workers. According to the Labour Code in Poland, shift work is defined as the performance of work according to an adopted working time pattern whereby the time of work of individual employees shifts after a specified number of hours, days or weeks [80]. In the classification of shift work [81], one can distinguish between permanent shifts and rotational shift systems, including or excluding night work. In the current study, the shift schedule of the surveyed employees included night work. The percentage of employees working in shifts is constantly growing in Poland [82]. Currently, this rate is 30.7% of all employees in Poland (including 7.85% completing night work) and is higher than the EU average of 17.7% [83] and the United States average of approximately 20% [84]. Participants belonged to three occupational groups: (1) bus drivers (*n* = 500), production workers (*n* = 506) and nurses (*n* = 504). The number of shift and non-shift workers in each occupational group was similar and amounted to 53.2% to 46.8% for bus drivers, 50.6% to 49.4% for production workers and 52.8% to 47.2% for nurses, respectively. The study was conducted between September–November 2020. Out of 2000 distributed questionnaires, 1751 (88%) were returned and 1510 (76% of the original pool) were complete and subsequently used for data analysis. The analysed group consisted of 702 women (53.5%) and 808 men (46.5%), between 20 and 65 years of age (M = 42.72, SD = 10.64). Work experience ranged from 1 to 47 years (M = 19.47, SD = 10.73). A one-way between-subjects ANOVA test showed no significant differences in the distribution of age, *F*(2, 1501) = 2.53, *p* = 0.061, and the length of service in the three analysed occupational groups *F*(2, 1505) = 1.54, *p* = 0.215.

All participants were treated according to the Helsinki Declaration’s ethical guidelines and received a hard copy of the questionnaires along with a letter explaining the purpose of the study. Full confidentiality of data and anonymity were secured. Participants were asked to fill out the questionnaires and seal them in envelopes, which research assistants subsequently collected.

### 2.2. Measurement

*Work ability*. Polish version of work ability index (WAI) was used to measure this variable [43,85]. WAI consists of seven questions related to (1) subjective assessment of current work ability compared with a lifetime best (top form), (2) physical and mental effort capacity required in the current job (work ability in relation to job demands), (3) estimated work impairment due to disease, (4) own prognosis of work ability for two years, (5) mental resources, (6) number of diseases diagnosed by physicians and (7) the number of sick leave during the past year (12 months). The global index of WAI was used in the study.

*Positive and negative emotions*. The Polish version of job affect scale [86,87] was used to measure emotions at work. In this approach, affect at work refers to the frequency and intensity of experiencing positive and negative emotional states at work during the last two weeks. The tool consists of 20 terms describing affective states, of which 10 each refer to positive affect (e.g., *enthusiasm*, *relaxation*) and negative affect (e.g., *hostility*, *fear*). For each emotion, the intensity of a given emotion should be indicated on a seven-point scale during the last two weeks of work (1—very weak, 7—very strong).

*Job resources.* The variable was measured with the COPSOQ II subscales [88], in Polish version [89]. The aggregated indexes based on factor scores of job resources related to a task, interpersonal and leadership resources were used [31]. Task resources included four factors: influence at work (*Do you have any influence on what you do at work?*), possibilities for development (*Can you use your skills or expertise in your work?*), variation of work (*Is your work varied?*) and role clarity (*Do you know exactly which area is your responsibility?*). Interpersonal resources included three subscales: colleagues support (e.g., *How often are your colleagues willing to listen to your problems at work?*), social community at work (e.g., *Is there good cooperation between colleagues at work?*) and horizontal trust (e.g., *Do the employees in general trust each other?*). Leadership resources consisted of six scales: quality leadership (e.g., *To what extent would you say that your immediate superior is good at solving conflicts?*), superiors support (e.g., *How often do you get help and support from your nearest superior?*), organisational justice (e.g., *Is the work distributed fairly?*), rewards (e.g., *Is your work recognised and appreciated by the management?*), predictability (e.g., *Do you receive all the information you need in order to do your work well?*) and vertical trust (e.g., *Does the management withhold important information from the employees?*). Each subscale contained three or four items, with answers from one (“Always” or “To a very large extent”) to five (“Never/Hardly ever” or “To a very small extent”).

### 2.3. Analytical Procedure

The IBM SPSS 27 and AMOS 27 statistical packages were used for data analysis. The programs used for statistical analyses of the data were developed by International Business Machines Corporation (IBM), headquartered in New York, United States.

Prior to the verification of the main model, descriptive statistics and comparative analysis were calculated. In order to determine the factor accuracy and estimate the parameters of fit, a confirmatory factor analysis (CFA) of the tools used in the structure proposed by its authors was also carried out. Their factor structure was also checked in the case of aggregated indices of job resources (related to task, interpersonal and leadership resources). The CFA aimed to confirm that (1) four subscales (related to influence at work, possibilities for development, work variation and role clarity) load one factor, known as task resources; (2) six subscales (related to the quality of leadership, superior support, rewards, predictability, organisational justice and superior trust) load one factor known as leadership resources and (3) three subscales (related to co-workers support, social community at work and co-workers trust) load one factor known as interpersonal resources. In addition, the structure of positive affect (PA), negative affect (NA) and work ability (WA) dimensions, which constitute one factor, was also verified.

In the case of unacceptable measures of model fit to the data, the analysis of model parameters and modification indices was carried out in order to check the sources of the problem. Where it was possible to improve the parameters of the measurement model without interfering with the scale structure (e.g., without removing individual items), it was decided to introduce additional parameters to the model—i.e., covariances of measurement errors of the questionnaire items—and to reassess the parameters of fit for the measurement model.

At the beginning, the data were analysed for multivariate outliers with the Mahalanobis distance method, and when the result turned out to be statistically significant at the level of *p* = 0.001, it was eliminated from the analysis [90]. Moving on to the analysis of the confirmatory factor analysis (CFA) results, the Tucker-Lewis index (TLI), the comparative fit index (CFI), goodness of fit (GFI), root mean square error of approximation (RMSEA) and standardised root mean square residual (SRMR) were adopted as fitting parameters. The results for TLI, CFI and GFI ≥ 0.90 indicate an acceptable model fit [91,92]. For RMSEA, results up to 0.10 indicate an acceptable model fit [93]. In addition, 90% CI intervals were also applied. When analysing confidence intervals, it should be noted whether the interval upper limit is greater than 0.10 since this may indicate a worse fit [94]. The last fitting parameter analysed was SRMR, for which the fitting results <0.10 indicate an acceptable fitting [95]. In addition, the size of the factor loads was analysed. After estimating the measurement models of the questionnaires, the internal consistency of the subscales was analysed using the coefficient Omega, which is more adequate for the analysis of latent factors than coefficient Alpha. The alpha coefficients were provided additionally, as these are more frequently used in statistical analyses.

For the main part of the analysis, structural equation modelling (SEM) was applied. Task resources (TR), leadership resources (LR), interpersonal resources (IR), positive affect (PA), negative affect (NA) and work ability (WA) were introduced into the model. The following were tested: (1) direct effects of TR, LR and IR on work ability as well as (2) indirect effects of PA and NA on TR/LR/IR–WA links in the groups of shift and non-shift workers. When analysing the results of structural models, the expected cross-validation index (ECVI) was additionally applied, whose lower result indicates a better fit of the model, as well as the Akaike information criterion (AIC) and consistent Akaike information criterion (CAIC), which adjusts to a sample size, and the lower value suggests a better fit [96].

## 3. Results

In the first step, we tested whether the proposed indices of positive affect, negative affect, work ability, task, leadership and interpersonal resources had an assumed structure, and also if it was justified to create a general factor-based index for each construct. For example, we have examined whether TR has a hierarchical structure consisting of four factors and whether the statements included in these four factors have a sufficient empirical basis to be gathered into one general index. In order to do so, we conducted a series of first and second-order confirmatory factor analysis (CFA). In the case of PA, NA, and WA, the first-order analysis was carried out in respect of a single factor, and in the case of task TR, LR and IR, the second-order analysis, as well. The results of TLI, CFI and GFI for PA, NA and WA factors are above 0.90 (Table 1), and in most cases, even above 0.95, which is an indication of good fitting [93]. The results obtained in RMSEA are at an acceptable level, looking at both the single result and the confidence intervals (Table 1). The situation is similar in the case of SRMR (Table 1).

Moving on to the analysis of the TR, LR and IR results, the level of TLI, CFI and GFI is above 0.90; however, the values are slightly better in the case of the first-order analysis (Table 1). The results obtained in the RMSEA are within the range from 0.64 to 0.89 and the SRMR within the range from 0.0421 to 0.0902, which confirms an acceptable fitting; however, in the case of the interpersonal (IR) resources scale, the confidence interval for RMSEA slightly exceeds the value of 0.10, which suggests a problem with fitting, but due to the fact that all other fitting parameters for this scale are at an acceptable level, the application of the mentioned index in original form was decided. In summary, all indicators analysed in the study had acceptable fitting parameters, and the first and second-order confirmatory factor analysis carried out for TR, LR and IR confirmed a similar level of models fitting to the data, enabling the inclusion of indicators into general factors.

The skewness distribution and kurtosis parameters analysis determined the normality of variable distribution (Table 2). The results of skewness and kurtosis in each analysed variable, apart from age, are within the range from −1 to 1, which confirms the assumption about the normality of the variable distribution [94]. Table 2 also shows the results of the comparative analysis. The results show the homogeneity in relation to all positive aspects of work (i.e., TR, LR, IR, as well as PA and WA); however, the difference is statistically significant in the case of NA. A higher level of negative emotion occurs in non-shift workers.

The theoretical model presented in Figure 1 was verified with SEM using AMOS 27 software. The model was tested concurrently for two groups of workers. The comparability of effects obtained in both groups is ensured by, e.g., the same count, which is an important factor when estimating fitting parameters [93]. When verifying, a more general model was used, which was gradually limited to correct inclusion errors [97], and finally, the estimated models were included in Table 3 and the comparison of model estimation without limitations. In the case of both groups, the models are saturated, and after limitation, it turned out that in the group of non-shift workers, the best-fit model was the one limited by four paths (see Table 3, Figure 2), and in the group of shift workers, it turns out that the best is the model limited by three paths (see Table 3, Figure 3).

The difference in estimating models for groups of employees was significant. In the case of the saturated model for non-shift workers, it was possible to estimate the model, but it was impossible in the group of shift workers. Another difference was obtaining a model which was best fitted to the data. Moreover, in this case, it was possible to estimate several models in the group of non-shift workers; however, in accordance with practice [97], the model best fitted to the data was presented (Figure 2), which also shows a path between TR and NA to be compared to the group of shift workers. On the other hand, in the case of the group of shift workers, the model presented in Figure 3 was the best fit for the data. Focusing on the strength of the relationship between job resources and WA, it turned out that in both the shift and non-shift groups, only LR is directly related to WA (Table 4). In addition, LR significantly affects positive (PA) and negative (NA) emotions, while the relationship with PA in relation to NA is stronger in both groups. IR is not directly related to WA or PA but has been significantly related to NA. This relationship is stronger in the group of shift workers (Table 4). TR is not directly related to WA in any of the groups; however, in the group of shift workers, a significant positive relationship with PA, as well as in the group of non-shift workers a relationship with NA was demonstrated in Table 4. This relation between TR and NA in the non-shift group is at the level of statistical tendency, and it is only presented for comparison to the model for shift workers because it would not be current in the best-fit model for non-shift workers (Table 4). When analysing the relationship between TR and emotions in the group of non-shift workers, it is worth noting that it is significant only for PA. In the case of NA, it was not presented for the selected model since the model estimated through this path is a worse fit to the data than the presented model, and once it is estimated, the relationship between TR and NA is statistically insignificant (Table 4).

## 4. Discussion

Numerous studies support the negative consequences of the shift work schedule for physical [6,10,14] and mental [2,3,4] health. On the other hand, few studies have undertaken the mechanism of developing positive phenomena in shift work, e.g., work ability. The study aimed to determine job-related predictors of high work ability in shift and non-shift workers groups. First of all, the direct relations between the three job resources groups and work ability were tested. Based on the typology of Berthelsen et al. [31], we proposed a broad and comprehensive approach to job resources, including 13 resources related to the three domains at work: task, leadership and interpersonal relations. Apart from the direct relations, the indirect effects of positive and negative emotions on job resources—WA link—were checked. We compared which of the two processes mediated more strongly in the relations—the first related to increasing positive emotions, or the second related to lowering negative emotions.

Initial comparative analyses showed no differences between shift and non-shift workers in relation to all studied positive factors at work—i.e., task, leadership and interpersonal resources, positive emotions, as well as work ability. However, differences occured in the case of negative emotions. The obtained results turned out to be consistent with the results of some previous studies, in which no statistically significant differences were observed in the levels of work ability [50,54], as well as happiness, life satisfaction and quality of life [98] due to the work schedule. Thus, it can be concluded that while working in shifts may lead to a higher risk of mental problems—such as high levels of exhaustion [99], anxiety and depression [1,3]—it probably influences the positive aspects of wellbeing at work to a lesser extent.

Models with the three aggregated types of job resources as independent variables and two mediators (positive and negative emotions at work) were tested separately for the two groups—shift and non-shift workers. SEM has shown that the mechanisms of shaping work ability in these two groups of workers are quite similar. Only LR was related to WA in a direct way, regardless of the work schedule. TR and IR did not predict work ability in these groups. The obtained result is partially consistent with some previous studies (e.g., [24,25]). For example, a longitudinal study found that supervisory relations (but not interpersonal relations and task resources) are predictors of work ability assessed after ten years [24].

With regard to the indirect effects, both kinds of emotions mediate the positive impact of the three analysed job resources on work ability. Interestingly, however, the impact that individual job resources had was different. LR led to “good” work ability by both enhancing PA and weakening NA. IR reduced NA but did not relate to PA. TR, conversely, intensified PA but is not related to NA. High PA and low NA led to high WA by using two paths.

So, we can assume that when employees have a satisfied need of belonging, feel as a part of the social community at work, have trust in their co-workers and receive support from them, they probably experience less anger, hostility, fear or sadness in their workplace. However, when employees feel appreciated, treated fairly and supported in solving tasks by their superiors and when they have a sense of influence at work and development opportunities, they more often experience a sense of pride, happiness and enthusiasm. These two separate processes shape high work ability in the groups of shift and non-shift workers, which translates into better health and more effective functioning in the work environment. For nurses, this means better medical care for their patients. Several systematic reviews of the literature have found that poor physical and mental health in medical professionals is associated with less safe patient care [100,101]. In the case of bus drivers, a “good” physical and psychological state is associated with greater safety on the road [102,103], while in the group of production workers it means fewer mistakes at work and higher performance [104].

When comparing the size of the effects of the three job resources, it can be observed that TR have the lowest importance for the development of WA, their relations are insignificant for WA and they are the weakest for PA. Job factors associated with social relations in the organisation (both with colleagues and superiors) play a more significant role. The obtained results are consistent with the observable phenomenon of the growing importance of interpersonal relations in today’s world of work [105], as well as their role in contemporary job stress theories [106]. While TR was regarded as most important for motivational outcomes in the job characteristics theory, a shift in work-life research has emerged during recent years. Researchers emphasise that jobs, roles, tasks, and projects are embedded in interpersonal relationships, connections, and interactions in contemporary organisations. Thus, relations with superiors and co-workers are more pervasive and vital than in the past: most organisations use teams to complete work [107], such that employees carry out their tasks and responsibilities interdependently [108]. More and more often, work teams must closely cooperate with individuals and teams from various departments, fields and industries, as well as with entities outside the organisation [109]. Increasingly, the health and wellbeing of employees depend on the quality of relationships with other people, at work and outside of work [105].

This research is not without limitations. One such limitation is the fairly simplified way of understanding the positive and negative functions of emotions in our research. This is because we consider positive emotions ad hoc, namely we consider positive emotions to be any that lead to positive personal and organisational outcomes, while the ones that lead to negative outcomes are considered as negative [110]. From an evolutionary perspective, emphasising the social and adaptive importance of emotions, it is assumed that no emotion is universally “good” or “bad”, but its value is context-dependent [111]. For example, anger caused by feelings of injustice can have negative consequences, e.g., counterproductive work behaviour [36], but it can also induce remedial behaviour to address the wrongdoing, also referred to as “moral” anger in organisations [100]. One meta-analysis showed that shame could have positive effects as it strengthens motivation and triggers actions aimed at prosociality and self-improvement [112]. One meta-analysis showed that shame could have positive effects when it strengthens motivation and triggers actions aimed at prosociality and self-improvement [113]. With regard to positive emotions, although inducing gratitude may lead to prosocial behaviour, it may also have burdening effects on the helpers [114].

Furthermore, the interactional effects were not controlled, for example between different groups of job resources or between job resources and demographic characteristics. It is possible that employees having “good” relations with other co-workers and receiving strong support from their superiors simultaneously have a higher work ability. It also seems that the age criterion in the superior–subordinate dyads may be important. Young leaders may be more empowered in their job, which creates more positive than negative effects on subordinates. Older and more experienced leaders, however, may raise more objections or grievances for the job of their subordinates, which is likely to generate more negative effects. Therefore, if we identified the age cohort dyads of the respondents and classified them as old–old, old–young, young–old and young–young on the superior–subordinate relationship, the results would be probably different depending on how they are matched with each other. In addition, the individual differences in tolerance to shift work were not taken into consideration. A meta-analysis study indicates that low scores on morningness, languidity and neuroticism, and high scores on flexibility, extraversion and internal locus of control are related to higher shift work tolerance [115]. These mental dispositions of employees probably play a key role in work ability and the process of perceiving job resources and experiencing emotions.

Another limitation relates to the specificity of the occupational groups taken into account. These professions have job-contextual demands, e.g., emotional demands related to working with patients [116] and overload resulting from the COVID-19 pandemic [117] for nurses, cognitive load, fatigue, stress and accident risk resulting from the road situation [118,119,120] for bus drivers and monotony in work [121] or exposure to hazardous substances [122] for production workers. Therefore, other types of resources may play a significant role in these professions and, thus, work ability is shaped in a slightly different way. For this reason, the results should be generalised with vigilance to other (than studied) occupational groups. Another issue is that the research was conducted during the COVID-19 pandemic; hence, some responses, such as the assessment of current work ability compared with a lifetime best, the number of sick leave periods during the past year or the frequency of experiencing various emotions at work, may be biased by the specificity of the current situation. During a pandemic, the organisation of work, the methods of job performance and the level of job demands are different from traditional ones. For example, nurses face a great amount of unusual job-related stressors, including staff shortages, insufficient equipment, inadequate protection from contamination, risk of infection, work overload, social stigmatisation, isolation, lack of contact with their families, as well as lack of consistent information about the spread of the virus, its contagiousness and the effectiveness of ways of prevention [123,124]. In the case of bus drivers, the COVID-19 pandemic also changed the organisation of work, such as reducing the number of bus trips, shortening working time and shift time and introducing passenger limits in buses [125]. For some production workers, in turn, the time of the pandemic resulted in an increase in job insecurity related to the lockdown and compulsory leaves [126]. We assume that the abovementioned factors, related to the specificity of the pandemic, could have had a significant impact on the obtained research results.

The most important limitation of our research concerns the applied cross-sectional design of research. This type of research is criticised because different variables are measured simultaneously, so their assessment may be distorted by the “current” moods of the respondents. Moreover, it is difficult to conclude on a cause-and-effect relationship on the basis of a single measurement of variables made at one point in time [123]. Although it is emphasised that emotions represent the immediate response to work environment situations [36], which to some extent justifies measuring them at the same point in time as resources, the development of work ability has a long-term nature. In addition, work ability is probably characterised by high dynamics and develops as a result of the influence of both job resources and job demands. Therefore, in future longitudinal studies, it would be promising to capture the interaction of these two groups of job-related factors.

## 5. Conclusions

As for the contribution of our study, one of them is to show that general wellbeing at work, including having job resources, experiencing positive affect and work ability, is similar in shift and non-shift workers groups. The only observed difference concerns negative emotions at work, the frequency of which is higher in non-shift workers. Another important contribution is the investigation of potential mechanisms of WA development. Although the direct impact of job resources on WA has been poorly supported (only in relation to leadership), it turned out that job resources trigger an affective process of enhancing work ability, and through high positive and low negative emotions, in the long run, contribute to better health of shift and non-shift workers. Firstly, we found that the more autonomy, role clarity and possibilities for development (task resources), as well as the better leadership, higher trust, supervisor support and rewards (leadership resources), the more positive emotions employees felt and consequently the higher their work ability. Secondly, the better relations with co-workers (interpersonal resources), in turn, the lower negative emotions at the workplace and higher work ability. The research results, apart from the cognitive value, which is the verification of the three theoretical models (JD-R [17], COR [18] and B&B [19,20]) in one study, also have important practical implications for managers, HR specialists and group leaders. They show that by enriching job resources, it is possible not only to build work wellbeing of employees, but also improve their functioning in the work environment.

## Figures and Tables

**Figure 1 ijerph-18-07730-f001:**
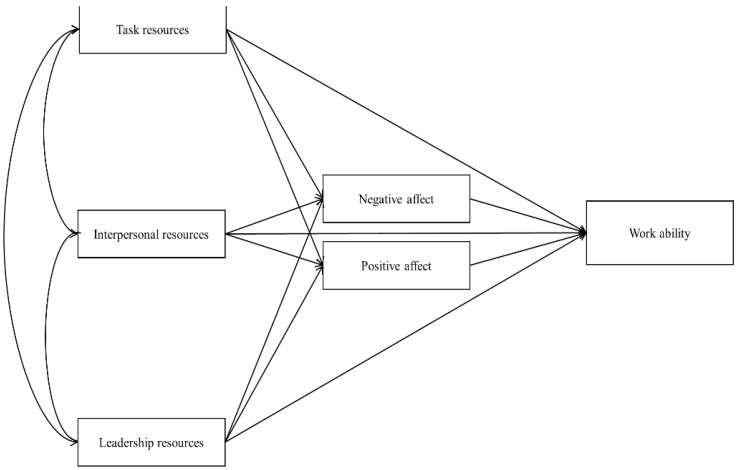
Theoretical model.

**Figure 2 ijerph-18-07730-f002:**
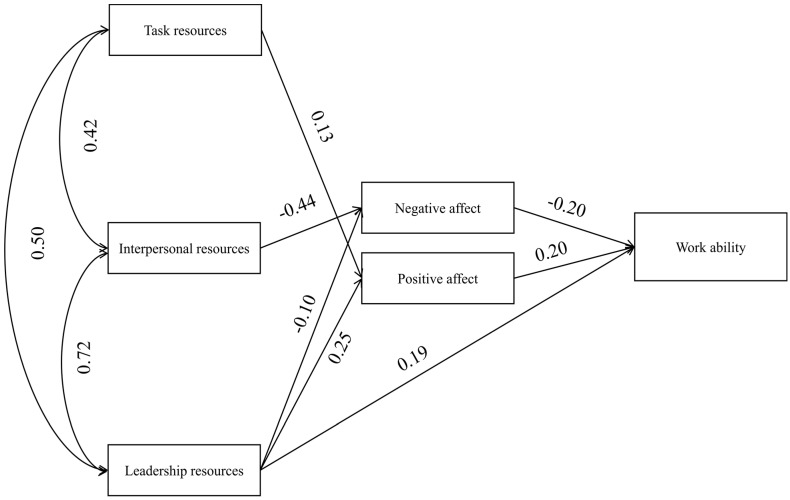
The final model for non-shift workers with standardised significant parameters consistent with the best-fit model.

**Figure 3 ijerph-18-07730-f003:**
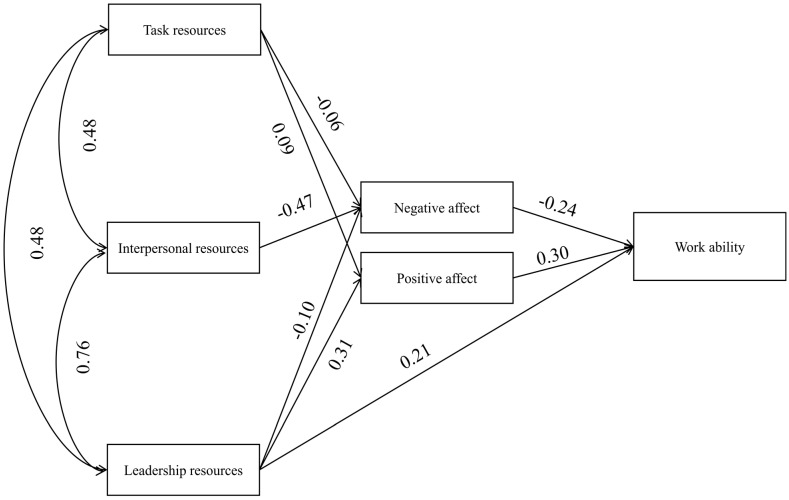
The final model for shift workers with standardised significant parameters consistent with the best-fit model.

**Table 1 ijerph-18-07730-t001:** Model adequacy and goodness of fit indices of the models tested using first- and second-order confirmatory factor analysis.

Models	Χ^2^	df	RMSEA	PClose	90 LLCI	90 ULCI	SRMR	TLI	GFI	CFI
(TR) first order	341.646	57	0.082	0.000	0.074	0.091	0.0810	0.911	0.932	0.935
(TR) second order	353.977	59	0.082	0.000	0.074	0.091	0.0856	0.911	0.929	0.933
(IR) first order	148.483	22	0.089	0.000	0.076	0.103	0.0902	0.949	0.958	0.969
(IR) second order	148.483	22	0.089	0.000	0.076	0.103	0.0902	0.949	0.958	0.969
(LR) first order	614.326	154	0.064	0.000	0.059	0.070	0.0421	0.933	0.919	0.946
(LR) second order	762.444	161	0.072	0.000	0.067	0.077	0.0587	0.917	0.904	0.929
(WA) first order	68.193	13	0.077	0.006	0.060	0.096	0.0587	0.951	0.974	0.970
(PA) first order	89.631	24	0.063	0.057	0.049	0.077	0.0363	0.952	0.973	0.968
(NA) first order	213.813	33	0.087	0.000	0.076	0.098	0.0295	0.955	0.944	0.967

Note. (TR)—task, (IR)—interpersonal, (LR)—leadership, (WA)—work ability, (PA)—positive affect, (NA)—negative affect, CFA—RMSEA root mean square error of approximation, PClose—*p* of close fit, 90 LLCI—confidence intervals, ULCI 90—confidence intervals, SRMR—standardised root mean square residual, TLI—Tucker-Lewis index, GFI—goodness of fit, CFI—the comparative fit index. The re-specifications of models were achieved based on error covariance modification indices.

**Table 2 ijerph-18-07730-t002:** Descriptive statistics and comparative analysis in non-shift group vs. shift group.

	Non-Shift				Shift				t	*p*	d-Cohen
	M	SD	SKE	KUR	M	SD	SKE	KUR			
Age	43.02	10.73	0.106	−1.07	42.41	10.54	0.086	−1.05	1.102	0.270	0.057
(WA) Work ability index	29.25	5.47	−0.710	0.110	29.20	5.47	−0.803	0.382	0.173	0.863	0.009
(TR) Task resources	55.52	13.76	0.213	−0.346	56.68	14.64	−0.246	−0.082	1.598	0.110	0.082
(IR) Interpersonal resources	60.55	17.35	0.089	−0.540	61.75	17.82	−0.307	−0.378	1.325	0.185	0.068
(LR) Leadership resources	60.75	15.02	0.157	−0.605	59.36	15.10	−0.024	−0.017	1.787	0.074	0.092
(PA) Positive affect	43.89	7.37	0.173	0.644	43.72	8.64	−0.008	0.186	0.413	0.680	0.021
(NA) Negative affect	28.49	11.27	0.125	−1.017	26.96	11.42	0.510	−0.297	2.628	0.009	0.135

Note. (WA)—work ability, (TR)—task, (IR)—interpersonal, (LR)—leadership, (PA)—positive affect, (NA)—negative affect, M—mean, SD—standard deviation, SKE—skewness, KUR—kurtosis, t—t-test result, *p* significance level, d-Cohen effect size.

**Table 3 ijerph-18-07730-t003:** List of model fitting values (1—sample model and 2—the best fitted model after paths have been removed).

	Non-Shift		Shift	
	Model 1	Model 2	Model 1	Model 2
Chi-kwadrat	2.084	5.498	6.551	8.341
Chi-kwadrat (*p*)	0.149	0.358	0.01	0.100
Cmin/df	2.084	1.100	6.551	2.085
TLI	0.986	0.998	0.946	0.989
GFI	0.999	0.997	0.997	0.996
AGFI	0.980	0.989	0.939	0.981
ECVI	0.058	0.052	0.062	0.057
LO 90	0.057	0.051	0.056	0.051
HI 90	0.070	0.066	0.078	0.073
CAIC	153.725	126.810	158.953	137.882
AIC	41.084	37.498	46.551	42.341
RMSEA	0.039	0.012	0.086	0.038
PCLOSE	0.462	0.923	0.117	0.653
90 LLCI	0.000	0.000	0.033	0.000
90 ULCI	0.115	0.054	0.154	0.075
SRMR	0.009	0.013	0.016	0.021

TLI—Tucker-Lewis index, GFI—goodness of fit, AGFI—adjusted goodness of fit, ECVI—expected cross-validation index, LO 90, HI 90—confidence intervals, CAIC—consistent Akaike information criterion, AIC—Akaike information criterion, RMSEA—root mean square error of approximation, PClose—*p* of close fit, 90 LLCI, 90 ULCI—confidence intervals, SRMR—standardised root mean square residual.

**Table 4 ijerph-18-07730-t004:** Standardised estimates with 95% bootstrap confidence intervals.

			Non-Shift				Shift			
Parameter			Estimate	95% LLCI	95% ULCI	*p*	Estimate	95% LLCI	95% ULCI	*p*
(PA) Positive affect	←	(TR) Task resources	0.134	0.036	0.222	0.003	0.087	0.006	0.163	0.033
(NA) Negative affect	←	(TR) Task resources	0.007 ^1^	−0.067 ^1^	0.081 ^1^	0.844 ^1^	−0.064	−0.134	0.004	0.065
(NA) Negative affect	←	(IR) Interpersonal resources	−0.441	−0.529	−0.347	<0.001	−0.472	−0.562	−0.383	<0.001
(PA) Positive affect	←	(LR) Leadership resources	0.247	0.166	0.337	<0.001	0.307	0.226	0.384	<0.001
(NA) Negative affect	←	(LR) Leadership resources	−0.101	−0.193	−0.008	0.032	−0.097	−0.185	−0.007	0.035
(WA) Work Ability Index	←	(PA) Positive affect	0.200	0.128	0.267	<0.001	0.296	0.229	0.361	<0.001
(WA) Work Ability Index	←	(NA) Negative affect	−0.204	−0.283	−0.122	<0.001	−0.243	−0.323	−0.168	<0.001
(WA) Work Ability Index	←	(LR) Leadership resources	0.186	0.106	0.264	<0.001	0.212	0.143	0.279	<0.001

^1^—the relationship provided for the purpose of comparing with the model evaluated with the task resources → negative emotions path characterised by worse fitting parameters. Standardised estimate, 95% LLCI, 95% ULCI confidence intervals, *p* significance level.

## Data Availability

The data sets generated for this study are available on request to the corresponding author.

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
