# Peer review of "How Is Work Ability Shaped in Groups of Shift and Non-Shift Workers? A Comprehensive Approach to Job Resources and Mediation Role of Emotions at Work"

_ijerph, 2021, doi:10.3390/ijerph18157730_

Round 1

Reviewer 1 Report

Dear Authors,

Thank you for your careful work and well-written manuscript.

The introduction is very well written and represents three theories that were combined to justify the main idea of this study. Next, the authors well document existing findings and gaps in the literature. 

Please check the sentence in lines 193-195: "For instance, some researchers showed that positive affect is a predictor of lower blood fat, blood pressure and a healthier body mass index [69], lower risk for heart disease [70] and self-rated health [71]." Did you mean better or improved self-rated health?

The study sample is well selected recruiting similar proportions of both genders and same occupational groups in shift and non-shift workers with no significant difference in workers age across them.

Please provide evidence of cultural adaptation of all measures used - work ability, positive and negative emotions and job resources and include them in the reference list. If I understand properly, the aim of your study was to test direct and mediated effects between work ability, emotions and job recourses while you start the results from a set of CFAs of measures used. Please revise your study aims.

Line 270: please check the SPSS AMOS version, I think the newest released version is the 26th.

Table 4. Please explain the abbreviations of the study variables presented. Also, it would be helpful for the reader to see standardized coefficients on the paths in Figures 2 and 3. Also, tested two separate models in shift and non-shift workers are very similar and path coefficients representing associations between study variables can not be compared. I would suggest additional multi-group analysis (shift vs. non-shift workers group) and models comparison according to measurement weights, structural covariances and residuals.

Also, PROCESS macro for SPSS by A. Haeys would allow testing moderated effects of occupational group and shift vs. non-shift workers group in mediation models. It might be helpful for your future research.

Thank you once again for the interesting paper. I will be happy to review it after revision.

Author Response

Dear Reviewer,

thank you very much for preparing your review and comments of our article. At the beginning, we wanted to point out that we made two additional independent language revisions to improve the clarity of the text. We address each comment below.

The introduction is very well written and represents three theories that were combined to justify the main idea of this study. Next, the authors well document existing findings and gaps in the literature. 

Please check the sentence in lines 193-195: "For instance, some researchers showed that positive affect is a predictor of lower blood fat, blood pressure and a healthier body mass index [69], lower risk for heart disease [70] and self-rated health [71]." Did you mean better or improved self-rated health?

Thank you for this comment. This sentence has been clarified - improved self-rated health was investigated in the cited study.

The study sample is well selected recruiting similar proportions of both genders and same occupational groups in shift and non-shift workers with no significant difference in workers age across them.

Thank you for this comment.

Please provide evidence of cultural adaptation of all measures used - work ability, positive and negative emotions and job resources and include them in the reference list.

All used measures (COPSOQ II, JAS and WAI)  have been adapted in Polish conditions. We have added Polish citations. The items have been also added to References section.

If I understand properly, the aim of your study was to test direct and mediated effects between work ability, emotions and job recourses while you start the results from a set of CFAs of measures used. Please revise your study aims.

In this paper, the description of the introduction to the statistical analyses was modified. The CFA presentation at the outset was intended to show the consistency of the dimensions that were used to estimate SEM in the next section.

Line 270: please check the SPSS AMOS version, I think the newest released version is the 26th.

Thank you very much for your vigilance, however we were already using spss package version 27.

Table 4. Please explain the abbreviations of the study variables presented.

All abbreviations have been explained, and new consistent notations have been added

Also, it would be helpful for the reader to see standardized coefficients on the paths in Figures 2 and 3.

In the original version of the file, figure 2 and figure 3 showed a standardized value next to each path and information about whether the path was significant. Formatting the file during upload made the values invisible

Also, tested two separate models in shift and non-shift workers are very similar and path coefficients representing associations between study variables can not be compared. I would suggest additional multi-group analysis (shift vs. non-shift workers group) and models comparison according to measurement weights, structural covariances and residuals.

Thank you very much for your suggestion, but our assumption of the SEM analysis was to test the theoretical model for the Shift vs. nonShift group and present which model for one group is the best fit to the data and which model for the other group is the best fit to the data. As shown in the paper there are differences in the best fitting model and also in the strength of the relationship between the variables in the different groups.

Also, PROCESS macro for SPSS by A. Haeys would allow testing moderated effects of occupational group and shift vs. non-shift workers group in mediation models. It might be helpful for your future research.

Thank you very much for this suggestion. We considered this solution, but in PROCESS macro there is a limitation to one independent variable, and we wanted to check three types of resources at the same time, and this prevailed in favor of SEM.

Thank you once again for the interesting paper. I will be happy to review it after revision.

Thank you very much for your important comments, the inclusion of which has certainly improved the quality of the text.

Reviewer 2 Report

Summary:

This article studies emotional empowerment on shift workers in Poland. Using 1,510 subjects working in the transportation, manufacturing, and healthcare industry, they find that leadership resources is a determining factor for both positive and negative emotions displayed at work, which, in turn, affected work ability and work performances.

General Comments:

According to lines 168 to 169, positive and negative effects are treat as distinctive but intercorrelated terms (thus, bi-dimensional on a Cartesian plane rather than unidimensional on a real line), so I wonder if it contradicts the saying in lines 65 to 66 that positive emotions are parenthesized as low level of negative emotions to a lesser extent, thus conceived as unipolar?

To make sure your definition is well understood and connected with the literature, can you explain what is shift and non-shift workers in your context?

Your survey is completed from Sep to Dec 2020. Do you expect some responses, such as the number of sick leave during the past year (12 months), are biased because of the COVID-19 pandemic?

You mentioned that task leadership resources (LR) is posting a statistically significant effect on both positive emotions (PA) and negative emotions (NA). Can you further explain how this connects with the moderate correlation with PA and NA as mentioned in line 170? In addition, I am expecting that leadership resources interact with age. Think of this, if the leader is young, then he may be more empowered in his job that poses positive more than negative effects on his or her subordinates. However, if the leader is old, then the opposite may hold true, especially he or she who thinks of himself or herself as more experienced, thus airing his or her grievances on the work ability of his or her subordinates. Therefore, if you further identify the age cohorts of the respondents, and classify them by old-old, old-young, young-old, and young-young superior-subordinate relationship, you may obtain different results depending on how they are matched with each other. Similarly, I am looking for any differences in across-group results among shift workers, such as bus drivers vs. production workers vs. nurses. Mention which of the results are affected by the clustering, and address them from the perspective of the distinctive work cultures.

Specific Comments:

Line 15. Should appear as “1,510”.

Line 17: Change stronger to strongly.

Line 21: You can remove the word “obtain”.

Line 30: You need to add the commas in between.

Line 36 to 41 is too long. Consider paraphrasing it.

Line 64. “the” most conductive.

Line 75. Last several -> past few

Line 78. Should you focus on international rather than just American journals?

Line 153. Can you change “higher” to “more intensive”, to be more precise?

Line 162. Can you move the connective “that” to between “is” and “quite”?

Line 210. Change the word check to either “investigate” or “evaluate”.

Line 211. Move the adjective “stronger” to before the phrase “work ability”

Line 211. I am hesitant to use the word “perhaps” because it connotes uncertainty.

Line 215. As I mentioned before, this should appear as “1,510”.

Line 217. You mentioned that shift workers are constantly growing all over the world? Can you give some background information to substantiate this? Ideally, one example should be from Europe, another one from North America, and the last one from other developing countries.

Line 220. You mentioned bus drivers and nurses, but in general, people working in the transport and healthcare industry work on shifts. Can you inform me of why you only narrowly include those two groups of people, rather than also considering truckers and doctors at large?

Line 228. Can you clarify if the subjects receive monetary incentives for filling out the survey?

Line 229. Can you explain why 75% completion is used as a benchmark for inclusion?

Line 301. Please define your abbreviation SRMR.

Line 315. Is there a reason you report CAIC instead of BIC?

Line 338. Unchanged -> its original

Line 346. Make sure the spaces appear consistently. After (LR), (IR), and (NA), as well as after the comma following interpersonal (before ability), you need to add (remove) the empty space. Besides, should either your model or this note rank IR before/after LR in a consistent order?

Line 351. Should the abbreviations mentioned in lines 308 to 309 be also used in Table 2? Also, for negative emotions, we typically report the real number 0.01 < x < 0.05 instead of grossly x < 0.05, unless you have a highly significant regression coefficient whose p-value is < 0.01.

Line 352. Can negative and positive effects be correlated with each other? Following lines 168 to 170, they are now “two relatively autonomous and unipolar dimensions, moderately correlated with each other”. So I am expecting a link, double-arrowed, between these two moderators.

Line 354. allowed determining -> determined

Line 360. significant statistically -> statistically significant

Line 366. Remove the word “further” which duplicates with gradually

Line 376. Too many empty spaces between Fit, and AGFI.

Line 381. The third arrow is biased upwards. Please fix it.

Line 285. In all figures, can you move the box positive effect upwards a little bit, so that it does not cross the arrowed-line connecting leadership and work ability? The same applies to all boxes.

Line 441. study -> studies

Line 448. did not relate -> is not related

Lines 450 to 455. This is a general description of your results. Can you give a real-life example in each of the industries you mentioned, (i) bus drivers, (ii) production workers, and (iii) nurses?

Line 462. been seen -> emerged

Line 472. Can you rewrite this paragraph emphasizing your two-fold contributions from the perspective of shift workers, i.e., how their stylized work cultures emphasize the empowerment from supervisors on subordinates’ work emotions to be able to succeed at work? This would more succinctly highlights your contribution, differentiating (and connecting) your research from (to) the mainstream literature which focuses on all (or non-shift) workers in general.

Lines 484 to 487. I think this description is far too general. I am expecting some more specific limitations such as reduced (prolonged) shifts in bus drivers (nurses) that may lead to amplified results in some of the measured items (e.g., negative emotions due to job uncertainty or risk of infection), but this is exactly how the pandemic is affecting shift workers at this moment of time.

Line 497. “were not taken into consideration”.

Lines 500 to 501. “relates to… that is taken into account”.

Line 508. You either remove the adjective or great or replace “great caution” with “vigilance”.

Line 519. Replace “good” with “promising”.

Lines 520 to 523. Please spell out all names in full, if possible.

Line 531. Do you have an expiry date (specific duration) for the IRB approval?

Author Response

Dear Reviewer,

thank you very much for preparing your review and comments of our article. At the beginning, we wanted to point out that we made two additional independent language revisions to improve the clarity of the text. Below, we refer to each of the comments in detail.

General Comments:

According to lines 168 to 169, positive and negative effects are treat as distinctive but intercorrelated terms (thus, bi-dimensional on a Cartesian plane rather than unidimensional on a real line), so I wonder if it contradicts the saying in lines 65 to 66 that positive emotions are parenthesized as low level of negative emotions to a lesser extent, thus conceived as unipolar?

We treat positive and negative emotions as two relatively separate but correlated phenomena. In fact, the entry in verses 65-66 could be misleading. It has been changed to be more consistent.

To make sure your definition is well understood and connected with the literature, can you explain what is shift and non-shift workers in your context?

We have clarified how shift workers are defined in Poland. According to the Labor Code in Poland, shift work is defined as the performance of work according to an adopted working time pattern whereby the time of work of individual employees shifts after a specified number of hours, days or weeks. Shift work can classified as permanent shifts, rotational shift systems, including night work or without night work. In our study, the shift schedule of the surveyed employees included night work.

Your survey is completed from Sep to Dec 2020. Do you expect some responses, such as the number of sick leave during the past year (12 months), are biased because of the COVID-19 pandemic?

Thank you for this comment. We fully agree with the Reviewer. Indeed, the research was conducted at a specific time - during the pandemic. Hence, some of the subjects' responses may be different from the time before the pandemic. This issue has been taken in the Limitations section . We have described the potential impact of the pandemic on job demands and work organization in the three occupational groups - nurses, production workers and bus drivers. Thanks to these modifications, the text is more related to the contemporary situation in the world.

You mentioned that task leadership resources (LR) is posting a statistically significant effect on both positive emotions (PA) and negative emotions (NA). Can you further explain how this connects with the moderate correlation with PA and NA as mentioned in line 170? In addition, I am expecting that leadership resources interact with age. Think of this, if the leader is young, then he may be more empowered in his job that poses positive more than negative effects on his or her subordinates. However, if the leader is old, then the opposite may hold true, especially he or she who thinks of himself or herself as more experienced, thus airing his or her grievances on the work ability of his or her subordinates. Therefore, if you further identify the age cohorts of the respondents, and classify them by old-old, old-young, young-old, and young-young superior-subordinate relationship, you may obtain different results depending on how they are matched with each other. Similarly, I am looking for any differences in across-group results among shift workers, such as bus drivers vs. production workers vs. nurses. Mention which of the results are affected by the clustering, and address them from the perspective of the distinctive work cultures.

We agree that independent variables can interact with each other (e.g. LR x IR or age x LR) in our study. The age criterion in the superior-subordinate dyads probably play the key role in assessment of leader by subordinates. Young/old leaders are probably different perceived by younger and older subordinates. We considered the lack of testing of moderational effects as limitation in our study and we included this part in the discussion.

The description of a moderate correlation between LR and PA/NA referred to analyses conducted on the whole group, whereas the statement" leadership resources (LR) is posting a statistically significant effect on both positive emotions (PA) and negative emotions (NA" refers to the SEM model, which is considered separately for the groups.

In addition, we have conducted a preliminary analysis of the differences between groups of employees and this will be the subject of another article in which we want to focus on the differences in perceived resource importance according to profession.

Specific Comments:

Line 15. Should appear as “1,510”. It has been done

Line 17: Change stronger to strongly. It has been done

Line 21: You can remove the word “obtain”. It has been done

Line 30: You need to add the commas in between. It has been done

Line 36 to 41 is too long. Consider paraphrasing it. It has been done

Line 64. “the” most conductive. It has been done

Line 75. Last several -> past few It has been done

Line 78. Should you focus on international rather than just American journals?

Unfortunately, we have not found a similar analysis for Europe.

Line 153. Can you change “higher” to “more intensive”, to be more precise? It has been done

Line 162. Can you move the connective “that” to between “is” and “quite”? It has been done

Line 210. Change the word check to either “investigate” or “evaluate”. It has been done

Line 211. Move the adjective “stronger” to before the phrase “work ability” It has been done

Line 211. I am hesitant to use the word “perhaps” because it connotes uncertainty. It has been done

Line 215. As I mentioned before, this should appear as “1,510”. It has been done

Line 217. You mentioned that shift workers are constantly growing all over the world? Can you give some background information to substantiate this? Ideally, one example should be from Europe, another one from North America, and the last one from other developing countries.

Data on the percentage of shift workers in UE and US have been added. The percentage of shift workers in Poland is 30.7% of all employees (including 7.85% night work) and is higher than the EU average (17.7%) and the United States (20%).

Line 220. You mentioned bus drivers and nurses, but in general, people working in the transport and healthcare industry work on shifts. Can you inform me of why you only narrowly include those two groups of people, rather than also considering truckers and doctors at large?

The presented research is part of a larger project on the physical and mental health of shift workers employed in these three specific occupational groups. These groups are particularly problematic in Poland. For example, according to the report Health at a Glance Europe 2020 prepared by the OECD, Poland has one of the lowest numbers of employed nurses per 1,000 inhabitants in Europe (5.1 compared to an average of 8.2 for EU countries and 17.7 for Norway, which is in the lead). The Polish Main Chamber of Nurses and Midwives (2017) forecast that by 2030 this ratio will have dropped to 3.81. The average age of Polish nurses, which is 52.2 years, is also worrying. Nearly 52% of all nurses are aged over 51 years old, while people up to 30 years old constitute only 5.5% of the workforce. Such a high average age is largely due to the emigration of younger, well-educated Polish nurses to Western European countries and little interest in studying nursing in Poland in recent years. The main problems of Polish bus drivers, in turn, are low wages and poor working conditions.

Line 228. Can you clarify if the subjects receive monetary incentives for filling out the survey?

The subjects did not receive financial rewards for taking part in the research.

Line 229. Can you explain why 75% completion is used as a benchmark for inclusion?

The description was revised and clarified because the procedure description was handled by someone other than the person who analyzed the data. Fully complete results were considered for the study. Thank you very much for catching this mistake.

Line 301. Please define your abbreviation SRMR.

It has been done

Line 315. Is there a reason you report CAIC instead of BIC?

CAIC is somewhere between AIC and BIC, because  BIC more often shows a "penalty" due to model complexity, and we wanted to keep as many paths as possible in the proposed models, but in the other hand we  did not select the AIC that would be the most liberal.

Line 338. Unchanged -> its original It has been done

Line 346. Make sure the spaces appear consistently. After (LR), (IR), and (NA), as well as after the comma following interpersonal (before ability), you need to add (remove) the empty space. Besides, should either your model or this note rank IR before/after LR in a consistent order?  

It has been done

Line 351. Should the abbreviations mentioned in lines 308 to 309 be also used in Table 2? Also, for negative emotions, we typically report the real number 0.01 < x < 0.05 instead of grossly x < 0.05, unless you have a highly significant regression coefficient whose p-value is < 0.01.

Thank you very much for your detailed comment. The significance level has been changed, as well as the table descriptions and also the captions under each table.

Line 352. Can negative and positive effects be correlated with each other? Following lines 168 to 170, they are now “two relatively autonomous and unipolar dimensions, moderately correlated with each other”. So I am expecting a link, double-arrowed, between these two moderators.

For positive and negative emotions, we considered placing a double line between them, but the relationship between positive and negative affect was insignificant in each model and we dropped this connection in order not to further complicate the model.

Line 354. allowed determining -> determined

It has been done

Line 360. significant statistically -> statistically significant

It has been done

Line 366. Remove the word “further” which duplicates with gradually

It has been done

Line 376. Too many empty spaces between Fit, and AGFI.

It has been done

Line 381. The third arrow is biased upwards. Please fix it.

It has been done

Line 285. In all figures, can you move the box positive effect upwards a little bit, so that it does not cross the arrowed-line connecting leadership and work ability? The same applies to all boxes.

For tables and figures, unfortunately the proportions changed after uploading the file to the system, resulting in a number of mistakes. For the manuscript after corrections, we will pay special attention to the view of the file after uploading to the system.

Line 441. study -> studies It has been done

Line 448. did not relate -> is not related It has been done

Lines 450 to 455. This is a general description of your results. Can you give a real-life example in each of the industries you mentioned, (i) bus drivers, (ii) production workers, and (iii) nurses?

In line with the Reviewer's expectations, this part of the manuscript was expanded. The importance of the obtained results for the three professional groups was emphasized . We noted that job resources (via positive and negative emotions) shape high work ability, which translates into better professional functioning of nurses, bus drivers and production workers, but also into the related social benefits. Several references have been added that pertain to this issue.

Line 462. been seen -> emerged It has been done

Line 472. Can you rewrite this paragraph emphasizing your two-fold contributions from the perspective of shift workers, i.e., how their stylized work cultures emphasize the empowerment from supervisors on subordinates’ work emotions to be able to succeed at work? This would more succinctly highlights your contribution, differentiating (and connecting) your research from (to) the mainstream literature which focuses on all (or non-shift) workers in general.

The Conclusions section has been moved to the end of the article and preceded by the Limitations section. We have added the theoretical and practical implications of the research.

Lines 484 to 487. I think this description is far too general. I am expecting some more specific limitations such as reduced (prolonged) shifts in bus drivers (nurses) that may lead to amplified results in some of the measured items (e.g., negative emotions due to job uncertainty or risk of infection), but this is exactly how the pandemic is affecting shift workers at this moment of time.

We have detailed the Limitation sections. We've added several threads, such as: (1) no interaction effect control (between different job resources and between demographic characteristics and job resources); (2) lack of control over the influence of personality traits; (3) potential distortion of the results obtained, resulting from a pandemic; during this time, both the organization of work, the methods of job performance and level of job demands are different from the traditional ones.

Line 497. “were not taken into consideration”.

It has been done

Lines 500 to 501. “relates to… that is taken into account”.

It has been done

Line 508. You either remove the adjective or great or replace “great caution” with “vigilance”.

It has been done

Line 519. Replace “good” with “promising”.

It has been done

Lines 520 to 523. Please spell out all names in full, if possible.

It has been done

Line 531. Do you have an expiry date (specific duration) for the IRB approval?

The study design was positively assessed by the Ethics and Bioethics Committee of the Cardinal Stefan Wyszyński University in Warsaw. The consent does not specify an expiry date.

Reviewer 3 Report

This study developed on 1510 workers assesses the types of resources most strongly associated with the work ability of shift and non-shift workers through emotions at work evaluations. The manuscript contains three keywords, three figures, four tables, and one hundred eleven references. Overall, it is a correct and well-conducted article.
Although the manuscript is correctly planned, some sections are very extensive. For example, the introduction section is too long, more than the discussion one. I think the conclusion section is really a continuation of the discussion. The conclusions should highlight, briefly and concisely, the most relevant findings of the study.

Supplementary comments on different sections of the manuscript are also made.

Abstract
Page 1, line 15. Abbreviations and acronyms should be explained the first time they are used, e.g. “structural equation modelling (SEM)”.

Keywords
Page 1, line 23. The manuscript shows three keywords. For keywords, where possible, please use Medical Subject Headings Terms (MeSH Terms). None of them is a MeSH term. Some alternative MeSH terms proposed could be “shift work schedule” better than “shift work” or “emotions” rather than “emotions at work”.

Introduction
Reference numbers 79 and 80 do not appear cited in the text, although they are in the list of references. Please, include them in the text.

Materials and Methods
Page 6, line 270. Please, provide more information about statistical programmes (company, address, etc.).

Discussion and Conclusions
Please see the second paragraph of this report.

References
Total number of manuscript references: 111.
A correct and very complete section. The reference format is according to the journal’s guidelines.

Figures
The manuscript presents three figures with appropriate legends.

Tables
The manuscript shows four tables.
As you do in Table 1, in the other tables, you should include a footer to the table explaining all abbreviations. 
In the last three columns of table 4, please use a period decimal-separator instead of a comma decimal-separator.

Author Response

Dear Reviewer,

thank you very much for preparing your review and comments of our article. At the beginning, we wanted to point out that we made two additional independent language revisions to improve the clarity of the text. We address each comment below.

This study developed on 1510 workers assesses the types of resources most strongly associated with the work ability of shift and non-shift workers through emotions at work evaluations. The manuscript contains three keywords, three figures, four tables, and one hundred eleven references. Overall, it is a correct and well-conducted article.
Although the manuscript is correctly planned, some sections are very extensive. For example, the introduction section is too long, more than the discussion one. I think the conclusion section is really a continuation of the discussion. The conclusions should highlight, briefly and concisely, the most relevant findings of the study.

As suggested by the Reviewer, we expanded the discussion  to make it more proportional to the Introduction section. In particular, we have emphasized the importance of the obtained results for the three professional groups and developed the Limitation part. We've added to the Limitation several threads, such as: (1) no interaction effect control (between different job resources and between demographic characteristics and job resources); (2) lack of control over the influence of personality traits; (3) potential distortion of the results obtained, resulting from a pandemic; during this time, both the organization of work, the methods of job performance and level of job demands are different from the traditional ones. Due to the development of the section, some new references have been added. As expected by the Reviewer, the Limitation part has been moved to the general part of the discussion. In the Conclusion section, we focused only on the most important contributions of our study. We noted that job resources (via positive and negative emotions) shape high work ability, which translates into better professional functioning of nurses, bus drivers and production workers, but also into the related social benefits. Several references have been added that pertain to this issue.

Abstract
Page 1, line 15. Abbreviations and acronyms should be explained the first time they are used, e.g. “structural equation modelling (SEM)”.

Thank you for the suggestion. Abbreviations and acronyms have been explained.

Keywords
Page 1, line 23. The manuscript shows three keywords. For keywords, where possible, please use Medical Subject Headings Terms (MeSH Terms). None of them is a MeSH term. Some alternative MeSH terms proposed could be “shift work schedule” better than “shift work” or “emotions” rather than “emotions at work”.

It has been changed.

Introduction
Reference numbers 79 and 80 do not appear cited in the text, although they are in the list of references. Please, include them in the text.

Thank you for the comment. Missing citations have been completed.

Materials and Methods
Page 6, line 270. Please, provide more information about statistical programmes (company, address, etc.).

Exact details of the software developer and version of the statistical package are provided in the article

Discussion and Conclusions
Please see the second paragraph of this report.

The discussion section has been developed. In the revised version this part is more interesting and versatile, in our opinion.

References
Total number of manuscript references: 111.
A correct and very complete section. The reference format is according to the journal’s guidelines.

Figures
The manuscript presents three figures with appropriate legends.

Tables
The manuscript shows four tables.
As you do in Table 1, in the other tables, you should include a footer to the table explaining all abbreviations. 
In the last three columns of table 4, please use a period decimal-separator instead of a comma decimal-separator.

Thank you very much for pointing out the tables, as there were a number of errors when uploading the file to the system. Each table and figure was pasted into the article once again. Additional footnotes were added and also symbols were unified and commas were changed to periods, but in the original version of the table there were periods everywhere.

We will check the file carefully after uploading.

Reviewer 4 Report

The authors have presented a highly interesting and novel approach.  Nonetheless, the major issue is with the presentation of the information.

The authors right away stated the objective but later includes the theoretical background. In this case, since the initial introduction is teeny-weeny information, the objectives seems to be off place. The introduction needs to better structured so the reader comprehend the relevance of the paper. 

The methodology is hard to follow, the authors need indicated clearer the procedure and process by which the data was obtained and later saved. The Ethical considerations should be placed independently and not in the middle of the 2.1. Participants and procedures.

The results include the Table 2 before the authors indicated in the text. This is an example again of how misplace the information is currently set on the manuscript. 

Besides, I would recommend to move the limitations from the conclusions to a subsection in the discussion. 

Author Response

Dear Reviewer,

thank you very much for preparing your review and comments of our article. At the beginning, we wanted to point out that we made two additional independent language revisions to improve the clarity of the text. We address each comment below.

Moderate English changes required

The authors have presented a highly interesting and novel approach.  Nonetheless, the major issue is with the presentation of the information.

The authors right away stated the objective but later includes the theoretical background. In this case, since the initial introduction is teeny-weeny information, the objectives seems to be off place. The introduction needs to better structured so the reader comprehend the relevance of the paper. 

As expected by the Reviewer, we have removed the objective of our research from the first part of the Introduction sections. In the revised version of the text, the objective appears after the description of theoretical models - JDR, COR and B&B. For a better understanding of the further discussion, we have inserted a generally formulated question, which we are trying to find an answer to in the research.

The methodology is hard to follow, the authors need indicated clearer the procedure and process by which the data was obtained and later saved. The Ethical considerations should be placed independently and not in the middle of the 2.1. Participants and procedures.

The results include the Table 2 before the authors indicated in the text. This is an example again of how misplace the information is currently set on the manuscript. 

Thank you very much for the above comments. The Ethical considerations have been moved to the end of the paragraph on Participants and procedure. On the other hand, the entire structure of the article has been arranged according to the journal guidelines. However, the layout of the tables and figures has been changed to make the text clearer, and the tables are in new locations.

Besides, I would recommend to move the limitations from the conclusions to a subsection in the discussion. 

As expected by the Reviewer, the Limitation part has been moved to the general part of the discussion. Additionally, we we've added to the Limitation several threads, such as: (1) no interaction effect control (between different job resources and between demographic characteristics and job resources); (2) lack of control over the influence of personality traits; (3) potential distortion of the results obtained, resulting from a pandemic; during this time, both the organization of work, the methods of job performance and level of job demands are different from the traditional ones. Due to the development of the section, some new references have been added. In the Conclusion section, we focused only on the most important contributions of our study. We noted that job resources (via positive and negative emotions) shape high work ability, which translates into better professional functioning of nurses, bus drivers and production workers, but also into the related social benefits. Several references have been added that pertain to this issue.

Round 2

Reviewer 1 Report

Dear Authors,

Thank you for the improved version of your manuscript, good job.

Author Response

Dear Authors,

Thank you for the improved version of your manuscript, good job.

Dear Reviewer,

We would like to thank you once again for your important comments and for accepting the revised manuscript.

Reviewer 2 Report

In Figure 1 and 2, "work ability" is of a different font; please fix it. (You may refer to Figure 3, where the fonts in all boxes are all consistently typed.) 

Lines 296 - 297. Grammatically, it should be "to a very large extent" and "to a very small extent". Please double check with your survey, and correct as deemed appropriate.

Line 361. Should be indication but not indicator. The subtle difference is that the statistic itself is an indicator, but the realized value of the statistic is an indication of good fitting.

Line 391. Simplify "which allows confirming the compliance with the assumption..." to "which confirms the assumption...".

Line 393. Can you replace "absence of differences" with "homogeneity"?

Line 410. You can remove "the case of" and say "...that in the group of non-shift workers,...", and this would then become consistent with line 411.

Line 412. You can remove "that is" because that has appeared right before.

Line 449. Should be "in both" rather than "both in" to be grammatically right.

Line 453. Can you rewrite this as "in the group of shift workers"?

Line 455. Can you clarify what the "relationship is", and justify what "statistical tendency" are you referring to.

Author Response

Dear Reviewer,

We would like to thank you once again for your important comments.

Below, we refer to each of the comments in detail.

In Figure 1 and 2, "work ability" is of a different font; please fix it. (You may refer to Figure 3, where the fonts in all boxes are all consistently typed.) 

It has been done. The names of all variables in the figures are written in Times New Roman  font and the font size is 12 points.

Lines 296 - 297. Grammatically, it should be "to a very large extent" and "to a very small extent". Please double check with your survey, and correct as deemed appropriate.

It has been done

Line 361. Should be indication but not indicator. The subtle difference is that the statistic itself is an indicator, but the realized value of the statistic is an indication of good fitting.

It has been done

Line 391. Simplify "which allows confirming the compliance with the assumption..." to "which confirms the assumption...".

It has been done

Line 393. Can you replace "absence of differences" with "homogeneity"?

It has been done

Line 410. You can remove "the case of" and say "...that in the group of non-shift workers,...", and this would then become consistent with line 411.

It has been done

Line 412. You can remove "that is" because that has appeared right before.

It has been done

Line 449. Should be "in both" rather than "both in" to be grammatically right.

It has been done

Line 453. Can you rewrite this as "in the group of shift workers"?

It has been done

Line 455. Can you clarify what the "relationship is", and justify what "statistical tendency" are you referring to.

It has been done. This sentence has been changed so that it is clear what it refers to and what is the point of presenting the relationship between TR task resources and NA negative affect in both groups.

Thank you so much for carefully reviewing our manuscript and pointing out items that needed to be revised.

We hope that the current version of the text is acceptable.

Reviewer 4 Report

First of all, I would like to congratulate the authors for the changes made. However, the issue continues to be with the presentation of the data. In this sense the table 4 is presented before the explanation in the manuscript, therefore, I would recommend to revise the results so the table 4 is after the last paragraph.

Author Response

First of all, I would like to congratulate the authors for the changes made. However, the issue continues to be with the presentation of the data. In this sense the table 4 is presented before the explanation in the manuscript, therefore, I would recommend to revise the results so the table 4 is after the last paragraph.

Dear Reviewer,

We would like to thank you once again for your important comments.

The structure of the tables and figures in the article you proposed is definitely more readable.

We hope that the current version of the text is acceptable.